# Role of DNA modifications in *Mycoplasma gallisepticum*

**Tatiana A. Semashko**[1,2]*, **Alexander A. Arzamasov**[1], **Daria V. Evsyutina**[1,2], **Irina A. Garanina**[1], **Daria S. Matyushkina**[1,2], **Valentina G. Ladygina**[1], **Olga V. Pobeguts**[1], **Gleb Y. Fisunov**[1,2], **Vadim M. Govorun**[2]

**1** Federal Research and Clinical Center of Physical-Chemical Medicine, Moscow, Russian Federation,
**2** Research Institute for Systems Biology and Medicine, Moscow, Russian Federation

* t.semashko@gmail.com

## Abstract

The epigenetics of bacteria, and bacteria with a reduced genome in particular, is of great interest, but is still poorly understood. *Mycoplasma gallisepticum*, a representative of the class Mollicutes, is an excellent model of a minimal cell because of its reduced genome size, lack of a cell wall, and primitive cell organization. In this study we investigated DNA modifications of the model object *Mycoplasma gallisepticum* and their roles. We identified DNA modifications and methylation motifs in *M. gallisepticum* S6 at the genome level using single molecule real time (SMRT) sequencing. Only the ANCNNNNCCT methylation motif was found in the *M. gallisepticum* S6 genome. The studied bacteria have one functional system for DNA modifications, the Type I restriction-modification (RM) system, MgaS6I. We characterized its activity, affinity, protection and epigenetic functions. We demonstrated the protective effects of this RM system. A common epigenetic signal for bacteria is the m6A modification we found, which can cause changes in DNA-protein interactions and affect the cell phenotype. Native methylation sites are underrepresented in promoter regions and located only near the -35 box of the promoter, which does not have a significant effect on gene expression in mycoplasmas. To study the epigenetics effect of m6A for genome-reduced bacteria, we constructed a series of *M. gallisepticum* strains expressing *EGFP* under promoters with the methylation motifs in their different elements. We demonstrated that m6A modifications of the promoter located only in the -10-box affected gene expression and downregulated the expression of the corresponding gene.

## Background

*Mycoplasma gallisepticum* belongs to the class Mollicutes, which is characterized by reduced genome size, lack of a cell wall, and simplified organization of cells. Owing to these properties, it serves as a convenient model for minimal cells [1–4]. Despite its simplicity, the overall regulation mechanisms in *M. gallisepticum* are poorly understood, and DNA modifications add another layer of complexity to the same.

The most frequent modifications of genomic DNA in prokaryotes include N6-methyladenine (m6A), N4-methylcytosine (m4C), and, more rarely, 5-methylcytosine (m5C) [5, 6].

**Data Availability Statement:** Sequencing data were deposited into the SRA with the accession no. SRX10666727 and link https://www.ncbi.nlm.nih.gov/sra/SRX10666727 for WT M. gallisepticum

and accession no. SRX10666728 and link https://www.ncbi.nlm.nih.gov/sra/SRX10666728 for the ΔS.MgaS6I mutant. Modified DNA data was deposited in the REBASE database (http://rebase.neb.com/cgi-bin/pacbioget?11602).

**Funding:** This work was supported by the Russian Science Foundation grant no. 19-74-10105 "The role of chromatin structure in minimal cell in the maintenance of housekeeping proteome homeostasis." The funders had no role in study design, data collection and analysis, decision to publish, or preparation of the manuscript.

**Competing interests:** The authors have declared that no competing interests exist.

**Abbreviations:** 2D-DIGE, two-dimensional difference gel electrophoresis; CV, coefficient of variation; m6A, N6-methyladenine; m4C, N4-methylcytosine; m5C, 5-methylcytosine; MRM, multiple reaction monitoring; MTase, methyltransferase; SMRT sequencing, single molecule real-time sequencing; RM system, restriction-modification system; TRD, target-recognition domain; TSS, transcription start sites; WT, wild-type; ΔS.MgaS6I, knockout mutant of *M. gallisepticum* in HsdS S.MgaS6I (GCW_02360).

These modifications are introduced by methyltransferases (MTases), which are generally components of the restriction-modification (RM) systems in bacteria. RM systems can be classified into four types. Type I RM systems form a multi-subunit complex consisting of MTase, restriction, and sequence-recognition subunits. Type II RM systems include two separate enzymes for modification and restriction. Type III RM systems form a complex between MTase and restrictase. Type IV RM systems consist of sole restrictases that, unlike the previous types, cleave modified DNA [7].

The primary function of RM systems is defense against exogenous DNA, such as phages and plasmids [7]. However, in prokaryotic organisms, orphan MTases of Type II RM systems without cognate restriction enzymes can mediate epigenetic regulation of gene expression through DNA modification. Examples of orphan m6A-MTases include DNA adenine MTase (Dam) in Gammaproteobacteria, cell cycle-regulated MTase (CcrM) in Alphaproteobacteria, and orphan m5C-MTase DNA cytosine MTase (Dcm) in *Escherichia coli*. Dam is known to participate in methylation-dependent bacterial gene silencing, DNA replication, and DNA mismatch repair, while CcrM participates in the regulation of the cell cycle [8]. Even though Dcm methylation is conserved, its role remains unclear [9]. A large number of other functional orphan MTases have been identified by means of high-throughput search for epigenetic modifications in prokaryotes [6]. In addition, complete RM systems can carry out both defense and regulatory functions. For example, it has been shown in *Helicobacter pylori* that methylation guided by the Type I RM system affects its gene expression [10] and switching to the variable Type III RM system may affect virulence [11]. For *Streptococcus pneumoniae*, switching between asymptomatic and invasive phenotypes is mediated by rearrangements in the Type I RM system [12].

RM systems in mycoplasmas are still under research. For example, a new Type II RM system for *Ureaplasma parvum*, isolated from human placenta, has recently been described [13]. Phase-variable systems are also present in mycoplasmas, in addition to classical RM systems. For instance, there are phase-variable forms of the Type I RM system in *Mycoplasma pulmonis*, where the polymorphic *hsdS* genes produced by gene rearrangement encode a family of functional S subunits with differing DNA sequence specificities [14]. A Type III RM system from *Mycoplasma mycoides* subsp. *capri* has been described, where MTase gene expression has a phase-variable activity [15]. DNA modifications in *Mycoplasma genitalium* and *Mycoplasma pneumoniae* have been characterized at the whole-genome level, and enzymes performing these modifications have been identified [16]. In a previous study, we characterized a transcription factor that controls the expression of the Type I RM system MgaS6I in *M. gallisepticum S6* [17].

In this study, we performed whole-genome identification of *M. gallisepticum S6* DNA modifications, analyzed the distribution of its motifs, and identified the enzymes responsible for the modifications. In addition, we also investigated the protective and epigenetic effects of the modifications on mycoplasma.

## Results

### Identification of genomic DNA modifications and distribution of their motifs along the genome

The whole-genome strand-specific detection of DNA modifications in *M. gallisepticum* S6 was studied using single-molecule real-time (SMRT) sequencing [18]. Currently, this technique is a major method for the detection of bacterial DNA modifications [6, 19]. SMRT sequencing allows for the identification of a number of modifications, one of which, m6A, has been predicted in *M. gallisepticum*. Mycoplasma genome was sequenced with 90× coverage per strand

**Table 1. Characteristics of the modification motif ANCNNNNCCT.** Mean Score–mean Modification QV of instances of this motif that were detected as modified; Mean IPD Ratio–mean interpulse duration (IPD) ratio of instances of this motif that were detected as modified; Mean Coverage–mean coverage of instances of this motif that were detected as modified.

| Motif | Position | Modification Type | Fraction | Number of Detected Motifs | Number of Motifs in Genome | Group of Motif | Partner of Motif | Mean Score | Mean IPD Ratio | Mean Coverage |
|---|---|---|---|---|---|---|---|---|---|---|
| ANCNNNNCCT | 0 | m6A | 1 | 1312 | 1312 | ANCNNNNCCT/ AGGNNNNNGNT | AGGNNNNNGNT | 139.5 | 5.96 | 89.8 |
| AGGNNNNNGNT | 0 | m6A | 0.97 | 1273 | 1312 | ANCNNNNCCT/ AGGNNNNNGNT | ANCNNNNCCT | 126.2 | 4.98 | 89.9 |

and with an average read length of 8 kb [S1 Table in S1 File, column wild-type (WT)]. The m6A modification was the only one identified with high quality. Modified positions with low-quality identifications also presented several motifs. Considering that these motifs were significantly degenerate and only a small fraction of them were modified in the genomic DNA, we concluded that they represented noise and only one actual DNA modification motif existed in the S6 strain. The predicted high-quality DNA modification motif was **A**NCNNNNCCT/ **A**GGNNNNNGNT (with the modified nucleotides indicated in bold). Methylation occurs at both the adenines of the double-stranded motif, in the direct and complementary strands. The characteristics of the modification motifs are shown in Table 1.

In total, 1312 sites of the modification motif were identified in the genome, with a medium density of approximately 1 site per 1 kb (Fig 1, rings B and C). There were several extended regions without modification sites, which were up to 6.7 kb long. Mobile genetic elements were predominant within the unmethylated regions. The coding sequences located within respective regions, apart from transposases, did not belong to any common functional group. However, the regions with the highest density of methylated sites corresponded to the gene clusters involved in virulence (S2 Table in S1 File).

Majority of the identified sites were fully modified at both the strands, with a median fraction of methylated adenines of 0.94 (Fig 1, ring B; Table 1). Nevertheless, there was a small fraction of hemi- and hypo-methylated sites. Only 39 sites were hemi-methylated. Since we performed a high-coverage SMRT sequence, we estimated the proportion of methylated adenines in each of the 1312 methylation sites, for adenines in both the strands. Fig 2 shows the number of sites at different methylation levels. Sites located in different parts of the genome relative to the origin and midpoint of replication, either on the plus or minus strand, were divided into different groups (Fig 2A). This corresponds to the maternal and daughter DNA strands during bacterial division, as well as, the leading and lagging strands during replication. We also compared the efficiency of methylation for both the strands, ANCNNNNCCT/ AGGNNNNNGNT, in the methylation motif (Fig 2B). The position of under-methylated adenines in the hemi-methylated sites did not depend on the location in the genome relative to the replichore (Fisher's exact test two-tailed P-value>0.05). Interestingly, all under-methylated adenines were located on the AGGNNNNNGNT strand of the motif (Fig 2, S3 Table in S1 File). There were only six hypo-methylated sites with a fraction of both modified adenines less than 0.75 (S4 Table in S1 File). Neither hypo- nor hemi-methylated sites featured additional sequence motifs, as seen in the fully methylated sites.

## RM systems in *M. gallisepticum* and their functionality

Among all known enzymes capable of modifying DNA, *M. gallisepticum* S6 has two annotated type I RM systems, both of which are predicted to methylate the N6-position in the adenine. The first system shares homology with the RM systems found in other strains of *M*.

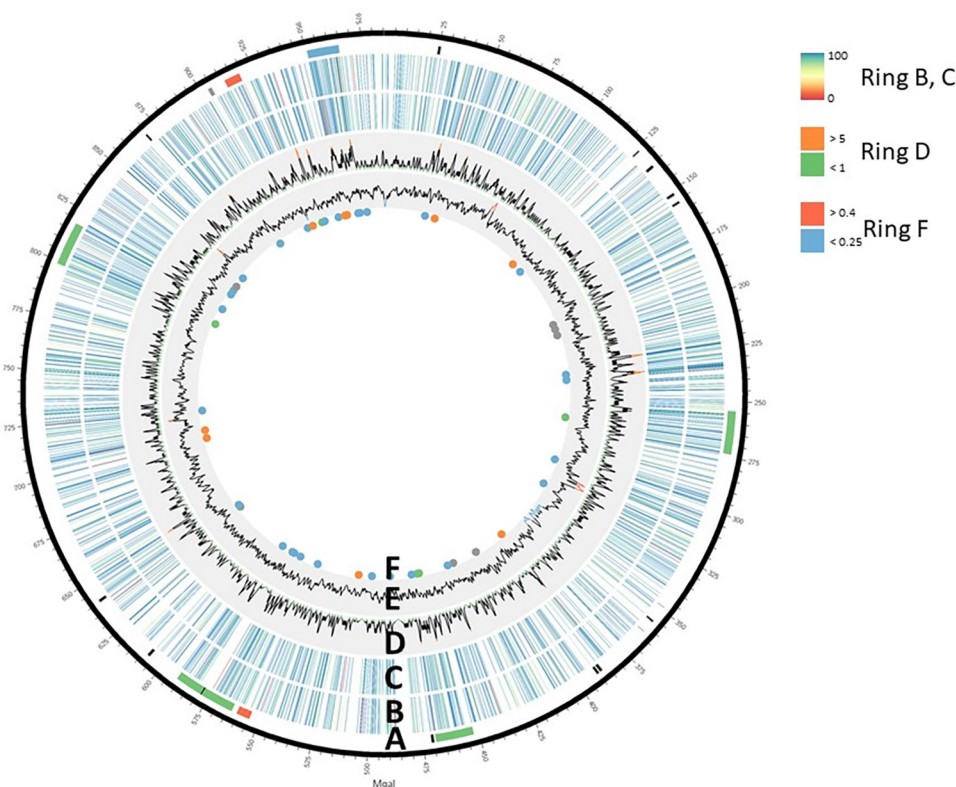

**Fig 1. Distribution of DNA modification sites along the genome of *M. gallisepticum* S6.** A–Annotated features: RM systems are in orange, *vlhA* clusters in green, the CRISPR system in grey, the virulence cluster in blue, and mobile elements in black. B and  –Whole-genome representation of the DNA modification motif ANCNNNNNCCT for plus and minus strand, respectively. The lines indicate positions of sites, and the colors from red to blue represent percentage of methylation in each site. Percentage of methylation was calculated using the build-in SMRT protocol, as the fraction of reads aligning to the position that has a modified base. D–Density of methylation sites. Maximum density of methylation is highlighted in orange, while minimum is highlighted in green. E–GC content. Maximum GC content is highlighted in red, while minimum is highlighted in blue. F–Hypo-methylated sites are in green, hemi-methylated sites in orange and blue (for plus and minus strands, respectively); proteins with differential abundance (as assessed using 2D-DIGE) between WT and ΔS.MgaS6I strains are in grey.

*gallisepticum.* Frameshift mutations have disrupted the genes encoding subunits of MTase HsdM and the DNA sequence-recognizing protein HsdS (specificity subunit). Mutation in the MTase is found only in the S6 strain, while disruption of the S-subunit is shared by all strains of *M. gallisepticum* we analyzed. The other RM system (MgaS6I) is intact, potentially functional, and unique for the S6 strain. MgaS6I is present in the S6 strain and none of the others, its representation in the genomic context of different strains is shown in S1 Fig. In addition to the proteins forming the RM system complex, it has a regulatory C-subunit and an HTH-family transcription factor, whose activity was demonstrated in our previous work [17] (Fig 3). Despite the frameshifts discussed above, transcripts and proteins of both the RM systems exist on the transcriptome and proteome levels of the S6 strain [3, 20].

To assign the identified methylated sites to the activity of a particular RM system, we obtained a knockout mutant with a disrupted sequence-recognition subunit S.MgaS6I (GCW_02360) of the unique RM system (ΔS.MgaS6I) using transposon mutagenesis. The Type I RM system works as a multi-subunit complex where the S subunit recognizes and binds to the DNA sequence, following which the MTase and restrictase are able to work. Therefore, we assumed that in mutants with both the S subunits disrupted, the DNA modification systems

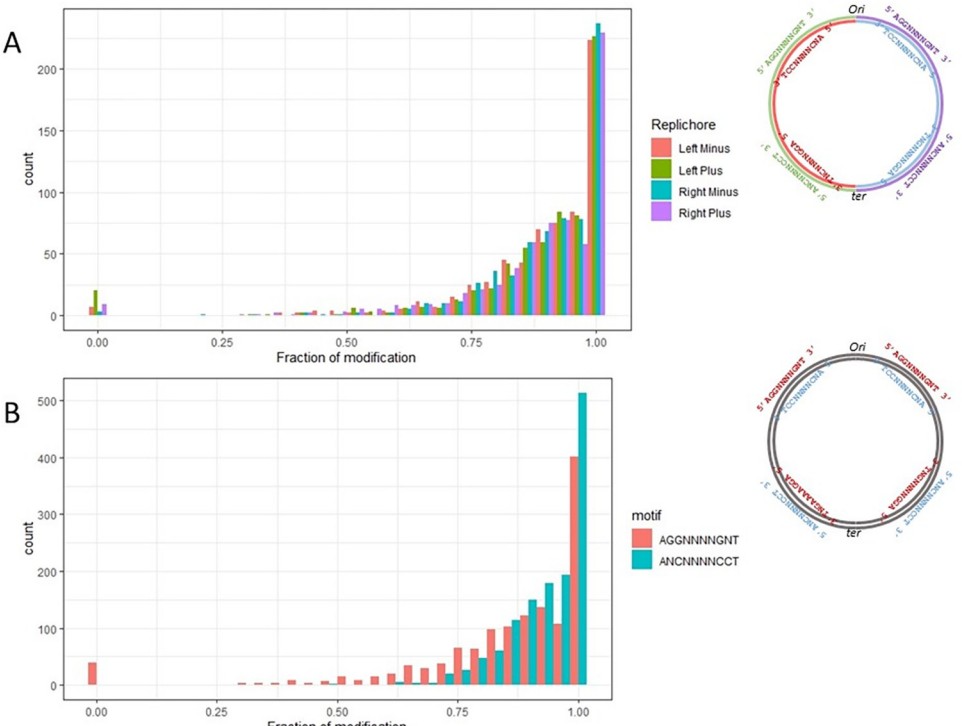

**Fig 2. Methylated fraction of each methylation site.** A–Number of sites with different fractions of methylation. Colors represent sites located in one of the four parts of the genome: between the origin and midpoint of the chromosome, on the plus or minus strand. The panel on the right schematically shows the methylation sites that were counted in each group. B–Number of sites with different fractions of methylation. In this figure, the location of sites in the genome has not been considered. Colors represent the sites on the ANCNNNNCCT or AGGNNNNNGNT strands in the double-stranded methylation motif. The panel on the right schematically shows the methylation sites that were counted in each group.

would be non-functional. Whole-genome DNA modifications in WT *M. gallisepticum* were compared with the modifications in ΔS.MgaS6I (S1 Table in S1 File). After *de novo* genome assembly, we did not observe any significant differences between the strains, except for the transposon insertion site (S7 Table in S1 File). The m6A modification and methylated ANCNNNNCCT motif were identified only in the WT strain, and the ΔS.MgaS6I strain lacked them. This indicates that the RM system common for *M. gallisepticum* strains is non-functional, at least in the S6 strain. The identified motif was typical for the Type I RM system [21] and belonged to the unique RM system MgaS6I. Proteins of this system are encoded by an operon consisting of four genes: XRE family transcriptional regulator (GCW_02350), methyltransferase (GCW_02355), specificity subunit (GCW_02360), and restrictase (GCW_02365).

MgaS6I was not found to have close homologs in other whole-genome sequenced *M. gallisepticum* strains. However, its particular components *hsdM* and *hsdS* demonstrated approximately 70% sequence similarity with the respective genes from *Enterococcus cecorum* strains SA1 and SA2, which are pathogenic isolates from chickens [22]. Within mycoplasmas, *hsdS* homolog was found in *Mycoplasma hyosynoviae* isolates from pigs with arthritis symptoms and shared about 65% sequence similarity [23].

The identified modification motif ANCNNNNCCT is unique among those represented in the REBASE database, and its specificity for HsdS homologs is unknown [7]. DNA binding of the HsdS subunits is mediated by the target recognition domains (TRDs) that define specificity of Type I RM systems. TRDs are two similar globular domains, each of which recognizes half

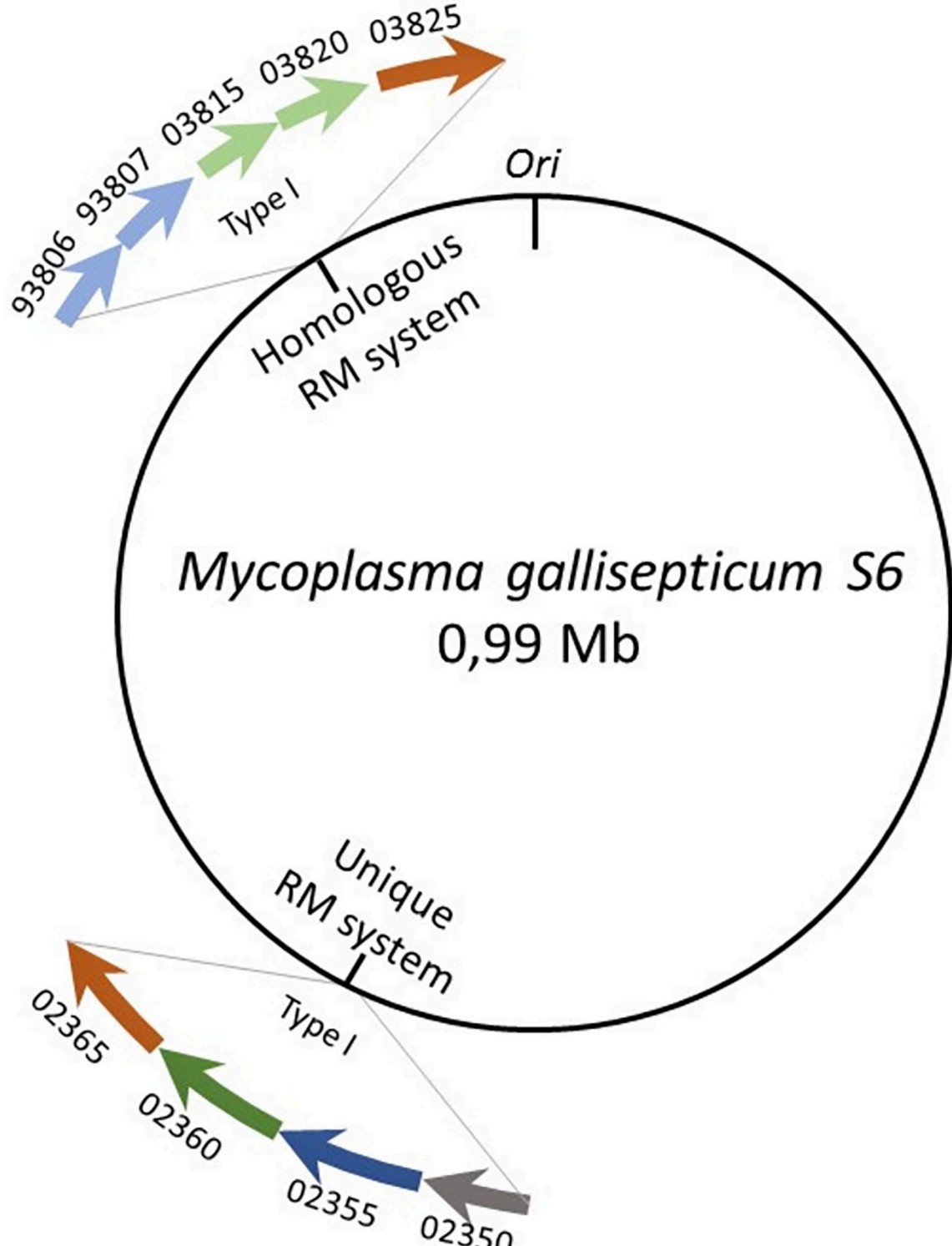

**Fig 3. Restriction-modification systems of *M. gallisepticum* S6.** Arrows represent the ORFs; grey, blue, green, and orange colors indicate genes *hsdC* (controller protein), *hsdM* (methyltransferase), *hsdS* (specificity subunit), and *hsdR* (restriction subunit), respectively. Ori–origin of replication. Faded colors represent disrupted genes.

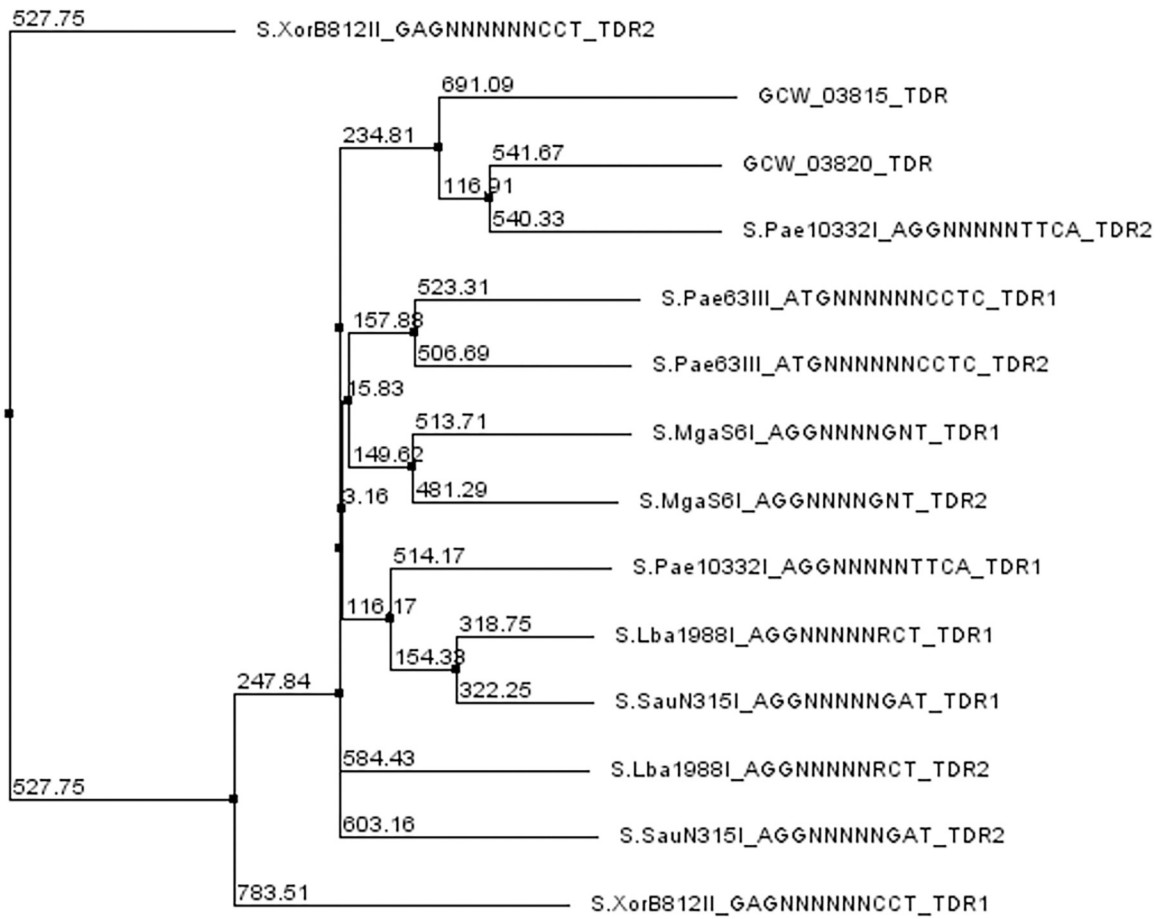

**Fig 4. Phylogenetic tree of TRDs of specificity subunits of *M. gallisepticum* S6 and other TRDs with similar specificity.** The names in the leaves represent the TRD names, which consist of the REBASE name of the HsdS and its sequence specificity. The numbers on the nodes are distance measures calculated using BLOSUM62 substitution matrices.

of the DNA sequence specificity motif [21, 24, 25]. TRDs recognizing AGG and ANC halves of the motif were selected using the REBASE database. Sequence comparison did not reveal a phylogenetic relationship between the TRD from MgaS6I and those from other S-subunits recognizing the same sub-motifs (Fig 4).

## Methylation effect of sites in the native location on gene expression

To investigate the effect of DNA methylation on gene expression, we searched for methylation sites in the regulatory regions of the genes using previous data on the transcription start sites (TSSs) in *M. gallisepticum* [3]. Only 26 genes have the methylation sites in their regulatory regions, those in the promoter (-40 to -1 relative to TSS), upstream to the promoter (-100 to -40 relative to TSS), or within the 5'-UTR (S5 Table in S1 File). The number of methylation sites in the regulatory regions of genes in *M. gallisepticum* S6 was underrepresented compared to the whole genome (as assessed using Fisher's exact two-tailed test P-value = 0.001). Only two modification sites overlapped with the core promoter of the genes of 50S ribosomal protein L31 (GCW_00775) and the molecular chaperone *dnaJ_4* (GCW_01610), both of which reside near the -35 box, which is insignificant for mycoplasma [3]. Several modification sites overlapped with putative regulatory sequences upstream of the TSS and 5'-UTR for 13 and 11

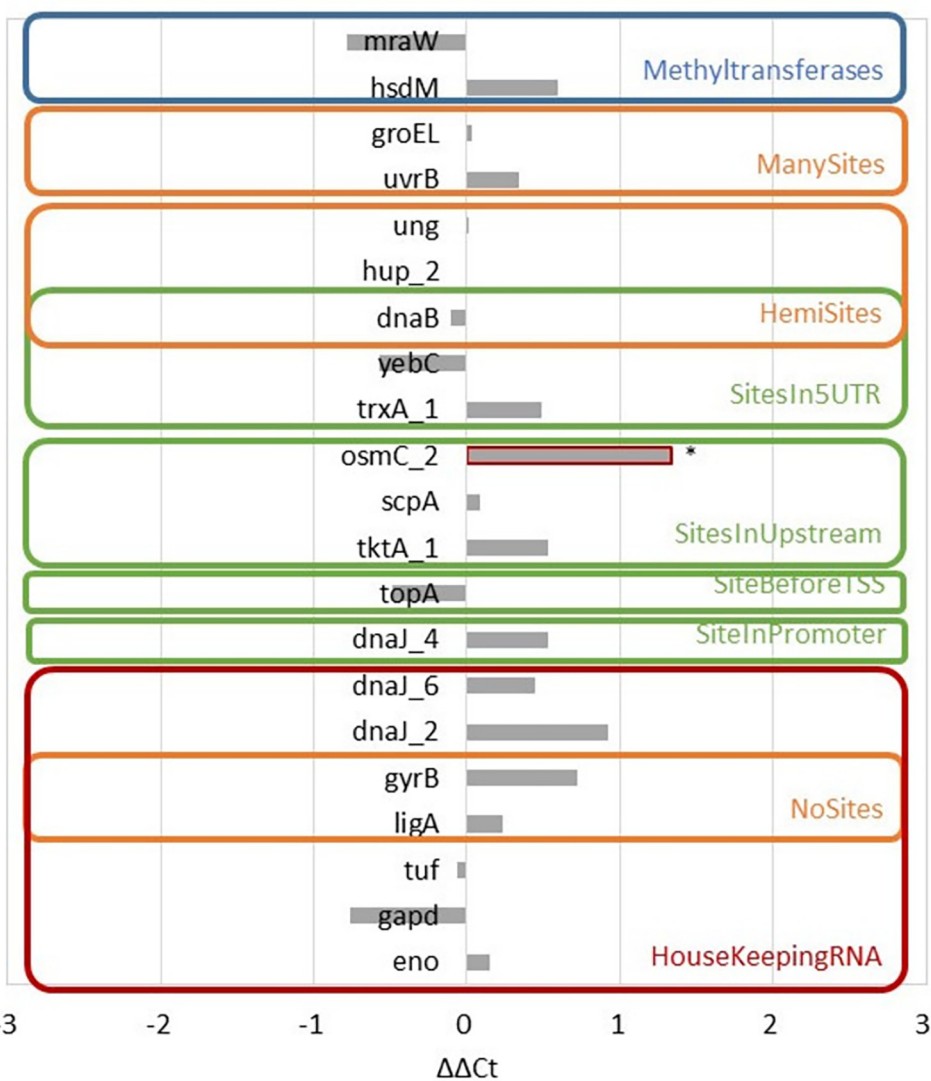

**Fig 5. Differences in expression levels of genes with DNA methylations sites between WT and ΔS.MgaS6I strains.**
Groups of control genes *eno*, *gadp*, *tuf*, *ligA*, *gyrB*, *dnaJ_2*, and *dnaJ_6*; genes with the methylation motif in promoters
*topA*, *dnaJ_4*, *tktA_1*, *scpA*, *osmC_2*, *trxA_1*, *yebC*, and *dnaB*; genes with different numbers of methylation sites or
with hemi-methylation status *ligA*, *gyrB*, *dnaB*, *hup_2*, *ung*, *uvrB*, and *groEL*; genes of methyltransferases *hsdM* from
MgaS6I and *mraW* have been presented. Genes with significant difference between strains (Student's *t*-test,
Benjamini–Hochberg correction, P<0.05) are shown using red boxes and asterisks.

genes, respectively. The RM system operon did not have modification motifs, in its own promoter or in the coding part of the controller protein.

The effects of methylation on gene expression were investigated using RT-qPCR by comparing gene expression in WT *M. gallisepticum* and the ΔS.MgaS6I strain (Fig 5). All strains were in the exponential phase of growth, similar to the conditions used for the SMRT analysis. The genes with a methylation motif in the -35 box of promoter (*dnaJ_4* and its homologs *dnaJ_2* and *dnaJ_6*, as additional controls), TSS (DNA topoisomerase I *topA* GCW_03565), upstream of the promoter (transketolase *tktA_1* GCW_00160, RNA 3-phosphate cyclase *scpA* GCW_92264, and peroxyredoxin osmC homolog *osmC_2* GCW_03005), within the 5'-UTR (thioredoxin *trxA_1* GCW_00460, transcriptional regulator *yebC* GCW_00725, and

replication initiation/membrane attachment protein *dnaB* GCW_01380) were selected as genes with methylation in potentially regulatory elements. All these genes, except *dnaB*, were found to be fully methylated in the SMRT analysis. Additionally, we studied the expression of genes with different numbers of methylation sites (without methylation sites–DNA ligase *ligA* GCW_03920 and DNA gyrase subunit B *gyrB* GCW_04070; genes containing more than average sites–excinuclease ABC subunit B *uvrB* GCW_00205 and molecular chaperone *groEL* GCW_02710) or with hemi-methylation status (sites in 5'-UTR–*dnaB* and uracil-DNA glycosylase *ung* GCW_03435; sites in protein–histone-like DNA-binding superfamily protein *hup_2* GCW_02335) (S3 and S4 Tables in S1 File). We also measured the gene expression of two methyltransferases, *hsdM* from MgaS6I and *mraW*. The genes encoding enolase *eno* GCW_02860, glyceraldehyde-3-phosphate dehydrogenase *gapd* GCW_01780, elongation factor Tu *tuf* GCW_01315, *gyrB*, and *ligA* were chosen as controls. Among them, *eno*, *gapd*, and *tuf* were considered as housekeeping genes, and the *gyrB* and *ligA* genes did not contain methylation sites in their sequence.

It was shown that there were no systemic effects on gene expression for methyltransferase genes, for the genes containing more than average sites or hemi-methylated sites or for the genes with the methylation sites in their regulatory regions. The last group was select genes with methylation sites within the 5'-UTR, genes with the methylation sites upstream to the promoter, gene *topA* with the methylation site near the TSS, and gene *dnaJ_4* with the methylation site in the -35 box of the promoter. These facts did not allow us to draw concrete conclusions about the potential influence of native methylation on gene expression.

## Protein changes in the absence of genomic DNA methylation

We further assessed the impact of DNA modifications at the proteome level by comparing the WT mycoplasma and ΔS.MgaS6I strain (Fig 6) using two-dimensional difference gel electrophoresis (2D-DIGE). Changes in the proteome can be caused by methylation and genetic modifications in the knockout ΔS.MgaS6I strain, which has a *tetM* insertion, and thus resistance to tetracycline. To discriminate these effects on protein abundance, we performed 2D-DIGE analysis for three other strains with *tetM* insertion in various genomic regions (2D-DIGE maps not shown). Changed only in ΔS.MgaS6I strain major proteins included single-stranded DNA-binding protein GCW_03460, phosphoglycerate kinase GCW_01785, cytadherence protein GCW_01845, and 4 proteins with unknown functions, GCW_00815, GCW_00850, GCW_00875, and GCW_02685. None of the changes observed were directly linked to the presence of DNA methylation in the respective genes, indicating that they were indirectly affected.

## Protective function of the RM system

The classic function of RM systems is to protect against exogenous DNA. To characterize this function of the MgaS6I system, we performed an experiment by introducing a transposon insertion into the ΔS.MgaS6I strain, which disrupted the RM systems in comparison to those of WT *M. gallisepticum*, and four other strains with *tetM* insertion in various intergenic regions. The transposon for this experiment carried resistance genes to chloramphenicol, *camR*, and five methylation sites ANCNNNNCCT, one of which is in *camR*. We demonstrated that inactivation of the MgaS6I RM system resulted in 19-fold more efficient transposon insertion (Fig 7).

## Effect of m6A modification in the promoter on gene expression

It is known that epigenetic signals that affect DNA-protein interactions are caused by methylated bases in DNA [26, 27]. In *M. gallisepticum* S6, methylation sites in the regulatory regions

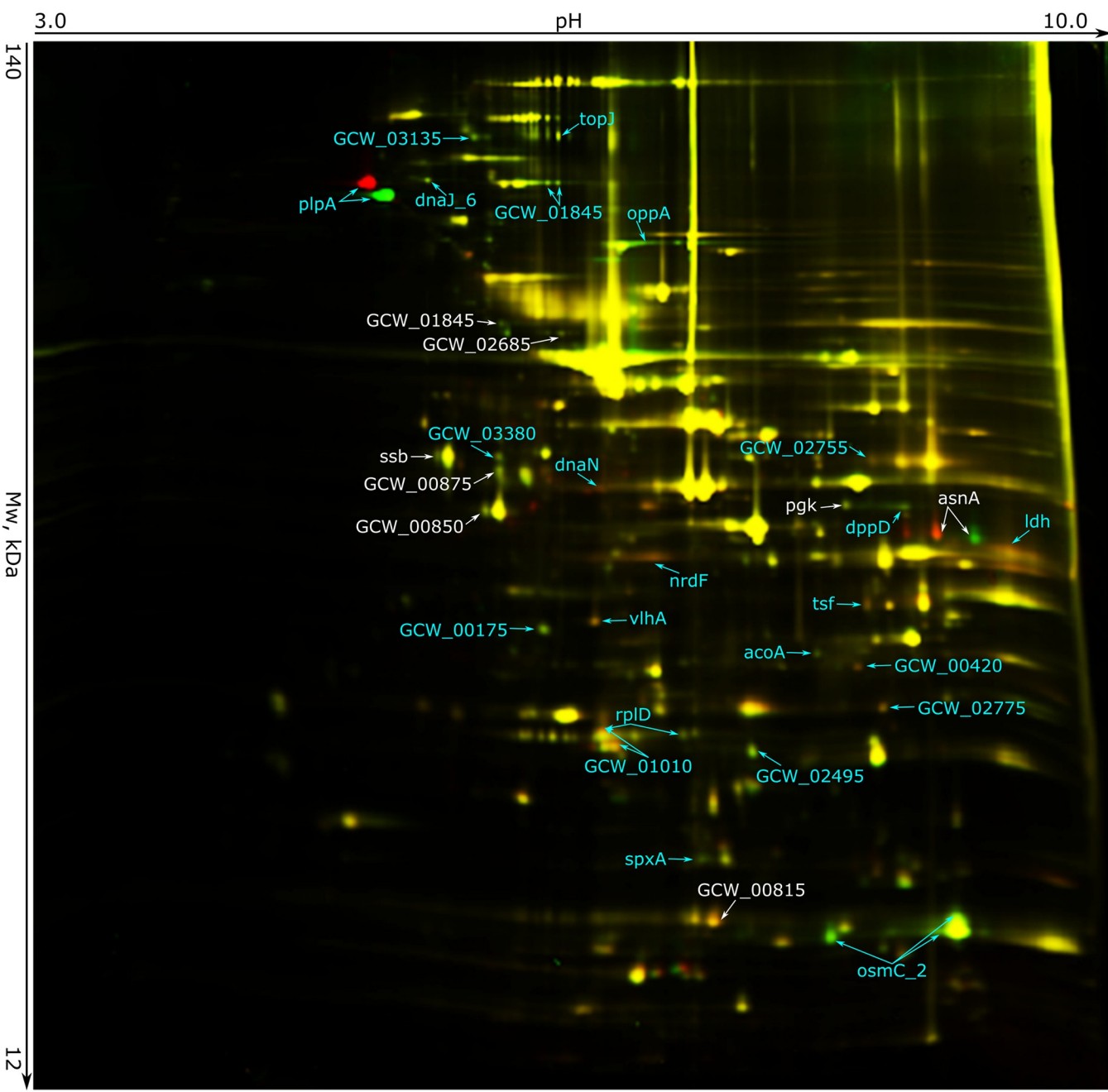

**Fig 6. Comparison of the proteomic profiles of the wild-type *M. gallisepticum* (red) and the ΔS.MgaS6I strain (green) with unmethylated DNA.** Proteins, whose abundances reproducibly changed in all the studied strains with *tetM* insertion in the various genomic regions, as compared to that in the WT, are in blue, while differences that are unique to ΔS.MgaS6I are in white.

are underrepresented, and no methylated base is located in the promoter sites important for gene expression, except for methylation in the -35 box, which is not very important for mycoplasma [3]. To investigate the effect of m6A DNA methylation on the efficiency of gene expression by RNA polymerase, we designed constructs with an *EGFP* reporter under different artificial promoters. Constructs included methylation sites in the promoters overlapping with -10 box, -35 box, region between the -10 and -35 box, or in the TSS; as a control, each construct had a corresponding construct, where the methylation site was inactivated by means of

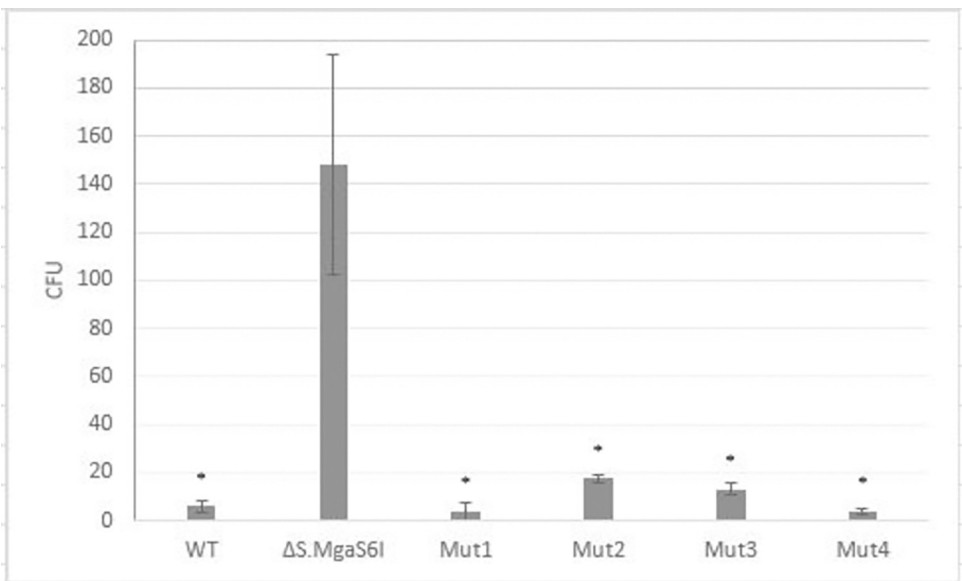

**Fig 7. The efficiency of *camR* insertion into wild-type *M. gallisepticum*, ΔS.MgaS6I strain, and four strains with *tetM* insertion in various intergenic regions.** CFUs of strains with significant difference, as compared to the CFU of ΔS.MgaS6I strain (Student's *t*-test, P<0.05), have been indicated using asterisks.

sequence inversion (Fig 8B). The inversion did not disrupt the promoter itself or the local GC content. Each of these constructs was integrated into the WT strain of *M. gallisepticum* using a transposon vector. Three biological repeats with different transposon locations in the genome for each promoter type were used for gene expression measurements. The genes *eno*, *gapd*, *tuf*, *gyrB*, and *ligA* were chosen as another set of controls, as done in the experiment on the methylation effect of sites in the native location on gene expression. We showed that the level of gene expression did not differ by more than two-fold between the paired constructs, with or without a methylated promoter, for almost all the studied genes. Only methylation of the -10-box resulted in a 4-fold downregulation of transcription (Student's *t*-test, Benjamini–Hochberg correction, P = 0.03) (Fig 8A).

## Discussion

In the present study, we investigated the DNA modifications in *Mycoplasma gallisepticum* and the enzymes involved in these modifications. We separately considered the modifying enzymes and the effect of the modifications on cell function.

We demonstrated that there is only one functional RM system, MgaS6I, in the *M. gallisepticum* strain S6, which seems to be acquired by *via* horizontal gene transfer, since it did not have close homologs in other whole-genome sequenced *M. gallisepticum* strains and a unique location in a genomic context compared to other strains. It is likely that for all these *M. gallisepticum* strains, the functional DNA methylation system exists only in the strain S6. MgaS6I is a Type 1 RM system. It differs from the more studied Type II RM systems as it functions as a multi-subunit complex in which the sequence recognition subunit HsdS is bound to DNA, and MTase and restrictase subunits are bound to HsdS. MgaS6I methylates the N6-position of adenine in the ANCNNNNCCT motif (methylation occurs at both the adenines of the double-stranded motif, in the direct and complementary strands).

In prokaryotes, there is an effect of underrepresentation in the genome of restriction sites of RM systems, which is called avoidance [28]. In this case, the recognition site is destroyed by

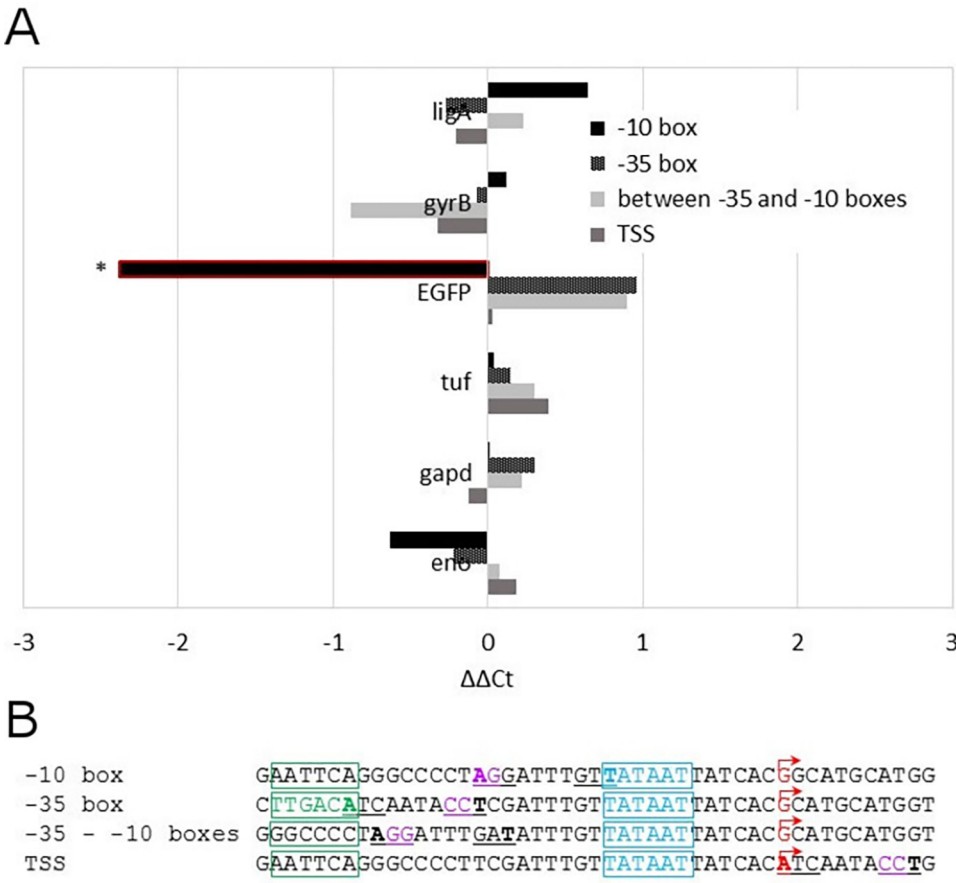

**Fig 8.** A–Difference between the expression levels of paired genes for mutants with *EGFP* genes under synthetic promoters, with or without methylation sites. For each pair, the difference for *EGFP* gene in comparison with control genes *eno*, *gadp*, *tuf*, *ligA*, and *gyrB* is shown. Genes with significant difference between promoters with or without the methylation site (Student's *t*-test, Benjamini–Hochberg correction, P<0.05) are shown with red boxes and asterisks. B–Sequences of paired *EGFP* promoters with or without methylated sites in -10 box, -35 box, between -35 and -10 boxes, and in the TSS. Methylation sites are underlined; methylated adenines in forward and reverse strands are in bold; -10 boxes are in blue and highlighted with blue boxes; -35 boxes are highlighted with green boxes and the sequence with strong consensus to the -35 box is in green; TSSs are in red and marked with red arrows; sequence differences between paired methylated or non-methylated promoters are in pink.

nucleotide substitutions in it. This phenomenon is mainly associated with Type II RM systems and depends on the lifespan of the RM system proteins in the organism[29]. No destroyed ANCNNNNCCT motifs were found in the S6 strain in comparison with other *M. gallisepticum* strains. This may be due to DNA cleavage at a distance from the recognition site by the Type I RM system. Another explanation is that since restrictase works in conjunction with MTase in the studied RM system of Type I, this imposes a limitation on the existence of unmethylated DNA sites and their further exclusion from the genome.

As mentioned above, almost all sites of the modification motif were fully methylated. Notably, only few methylation sites in genomic DNA were hypo- or hemi-methylated, and we did not identify any additional common sequence motifs in these cases. We hypothesize that hypo- or hemi-methylation may be caused by the lower local accessibility of DNA, particularly due to the chromatin structure or occupancy of nearby positions by DNA-binding proteins.

An interesting characteristic of MgaS6I is greater methylation efficiency for the ANCNNNNCCT strand, as compared to the AGGNNNNNGNT strand. Thus, the level of

methylation does not depend on whether the DNA chain is newly synthesized during replication but depends on MgaS6I affinity instead. The apparent affinity is stronger on the AGGNNNNNGNT strand of the methylation motif, taking into account that binding and methylation occur on opposite strands [21].

The main function of RM systems in bacteria is to protect against foreign DNA. Several mycoplasma DNA viruses have been described to date. The list includes dsDNA phages of *Mycoplasma bovirhinis*, *M. hyorhinis*, and *M. pylmonis* [30], which are characterized by their shape. Additionally, sequences of phages of *M. pylmonis* [31] and *M. arthridis* [32] were determined, and prophages in the genomic sequences of *M. fermentans* [33] and *M. agalactiae* [34] were also discovered. In the example of *M. gallisepticum* transformation, we demonstrated that the lack of MgaS6I RM system increases the transformation efficiency by 19-fold. Our data is consistent with that of another study in which the authors cloned a *Mycoplasma mycoides* genome as a yeast centromeric plasmid. Only considering the action of RM systems during genome transplantation into *Mycoplasma capricolum*, a sufficient number of viable *M. mycoides* cells were obtained in this study [35].

RM systems may also demonstrate the properties of selfish elements [36]. We found no evidence that MgaS6I enzymatic activity affects the Mycoplasma genome, but we have previously shown that its presence in the host genome has changed the implementation of genetic information. The controller subunit HsdC of MgaS6I can bind a set of promoters in addition to one in its own operon [17]. Thus, indirect participation of MgaS6I in the evolution of *M. gallisepticum* can be proposed.

There is growing evidence of the epigenetic role of m6A modifications in bacteria. So, N6-methylated adenine can influence DNA-protein interactions [27]. The number of methylation sites in the promoter regions of genes in *M. gallisepticum* S6 is underrepresented compared to the whole genome sequence. We also estimated enrichment ratio of ANCNNNNCCT in set of promoter sequences and in 100 sets of random generated sequences with the same length and GC content of the M. gallisepticum S6. In this case enrichment ratio was 0.49±0.11. Additionally, ANCNNNNCCT sites in the genome of the closest strain without the MgaS6I system, *M. gallisepticum* Rlow, exhibits a similar level of counts. Therefore, underrepresentation of methylation sites in promoter regions is possible due to the overall low GC content.

We did not find any influence of native methylation on gene expression in *M. gallisepticum* S6. In addition, phenotypic differences in the absence of methylation, as detected using 2D-DIGE, did not show clear correlations between methylation DNA and variation in protein levels. For *M. pneumoniae*, there was no correlation between the MTase M.MpnI gene and genes with methylation sites in their regulatory sequences at the transcription level [16]. We can speculate that *M. gallisepticum* S6 acquired the RM system MgaS6I, which did not introduce dramatic changes in cell regulation, as compared to that of the other strains of *M. gallisepticum*.

To demonstrate the possibility of epigenetic regulation by m6A in Mollicutes, we constructed *M. gallisepticum* strains with *EGFP* expression under artificial promoters with methylation motifs in their different elements, which were lacking native methylation sites in *M. gallisepticum* S6. We showed that only methylation in the -10 box of the promoter affected gene expression. Previously, we demonstrated that the -10 box and the first nucleotide of the transcript are the main features of functional promoters in *M. gallisepticum* [3]. We assume that methylation of the -10 box of the promoter results in weaker binding of the RNA polymerase complex or alters the melting or conformation of DNA strands in the promoter, thus leading to downregulation of expression of the respective gene. Another reason might be the competition between MgaS6I and RNA polymerase for the promoter. On extrapolating these data to all representatives of Mollicutes with a reduced genome and RNA polymerase with

close homology, we showed that m6A modification affects gene expression. Consequently, the presence of DNA modifications in the -10 box of promoters of genes of *M. gallisepticum* could play a role in the epigenetic regulation of *M. gallisepticum* and related bacteria.

## Conclusion

In this study, we identified DNA modifications in *M. gallisepticum* S6 at the whole-genome level. The only functional system capable of DNA modification was MgaS6I, a unique Type I RM system, encoded by genes- MTase GCW_02355, the specificity subunit GCW_02360, and the restrictase GCW_02365. We identified a unique methylation motif ANCNNNNCCT according to the REBASE database. Our results indicate that the MgaS6I RM system has a protective function and no regulatory effect in *M. gallisepticum*, that is, native methylation of DNA does not directly affect gene expression or protein abundance. This is due to the lack of methylation in the regulatory regions of the genes. When m6A modifications are present in the -10 box of the promoter, they influence gene expression and downregulate the expression of the corresponding gene in *M. gallisepticum* and related bacteria.

## Methods

### Cell culture

The *Mycoplasma gallisepticum S6* strain was provided by Prof. S.N. Borkhsenius, Institute of Cytology, St. Petersburg, Russian Academy of Science. *M. gallisepticum S6* was cultivated in a liquid medium containing tryptose (20 g/l), Tris (3 g/l), NaCl (5 g/l), KCl (5 g/l), yeast dialysate (5%), horse serum (10%), and glucose (1%) at pH = 7.4 and 37˚C under aerobic conditions. Cells were passaged at a dilution of 1:10, twice for 24 h, starting from the frozen culture prior to the experiment [37].

### Construction of transformants of *M. gallisepticum*

The knockout mutant in hsdS GCW_02360 (ΔS.MgaS6I) was selected from a library of random transposon knockouts (Arzamasov et al., manuscript in preparation). Construction of a vector for the transformation of *M. gallisepticum* was performed as described in [3]. Transformation was performed using electroporation, as described previously [38]. Localization of the position of transposon integration was confirmed using Sanger sequencing on an ABI Prism 3730XL system (Applied Biosystems, USA) using the primers W2 (5′ –AACCTGTATGC– CAACCGAGG–3′) and W3 (5′ –GTGAGCAAGTACACCGATGTTAAT–3′), complementary to the sequences of the ends of the tetracycline resistance gene *tetM*. Localization of the transposon position and its single insertion in the ΔS.MgaS6I mutant was additionally confirmed using SMRT sequencing.

For studying the effect of methylation on gene expression, strains with *EGFP* expression under different promoters were constructed, and the *EGFP* gene was cloned into the Tn4001-based vector described in [3], between the SphI and XbaI sites. Promoter variants were chemically synthesized and cloned into the plasmid, between the ApaI and SphI sites. Sequences of oligos for promoters with methylations in the following regions: -10 box, -35 box, between -10 and -35 box, and TSS are shown in S6 Table in S1 File. Transformation and localization of the transposon integration position were performed for the ΔS.MgaS6I mutant. Three biological samples were selected for the transformants with artificial promoters.

The mutants used as controls in the experiments involving 2D-DIGE and assessment of the protective function of the MgaS6I RM system were selected from the same library of random transposon knockouts as the ΔS.MgaS6I mutant.

## DNA isolation

DNA was isolated from WT *Mycoplasma gallisepticum S6* and the ΔS.MgaS6I mutant in the exponential phase of growth using the QIAamp® DNA Micro Kit (Qiagen, USA), according to the manufacturer's instructions. The amount of DNA was determined using a Qubit® 2.0 fluorometer (Thermo Fisher Scientific, USA), and the quality of DNA was estimated using electrophoresis on a 1% agarose gel.

## DNA sequencing

SMRT sequencing [18] was performed at the DNA Sequencing and Genomics Lab, Institute of Biotechnology, University of Helsinki. Libraries were prepared following the SMRTbell template preparation protocol with 8–10 kb inserts and sequenced on the PacBio® RS II (Pacific Biosciences, USA) using DNA/Polymerase Binding Kit P6 v2, DNA Sequencing Reagent Kit 4.0 v2 with C4 chemistry, and SMRT Cell v3. Each sample was sequenced in 1 SMRT cell. The coverage of the genome per strand was approximately 90× for WT *M. gallisepticum S6* and approximately 110× for the ΔS.MgaS6I strain (S1 Table in S1 File).

## Bioinformatics analysis

*De novo* genome assembly, modification detection, and motif search were performed using the SMRT® Analysis software. Sequencing data were deposited into the Sequence Read Archive (SRA) with the accession SRX10666727 and link https://www.ncbi.nlm.nih.gov/sra/SRX10666727 for WT *M. gallisepticum* and accession SRX10666728 and link https://www.ncbi.nlm.nih.gov/sra/SRX10666728 for the ΔS.MgaS6I mutant. The circular whole-genome sequence of *M. gallisepticum S6* was assembled by us earlier https://www.ncbi.nlm.nih.gov/genome/1113?genome_assembly_id=266085 [2]. Modification detection and motif analysis were performed using the protocol RS_Modification_and_Motif_Analysis.1. Modification data were deposited in the REBASE database [7] (http://rebase.neb.com/cgi-bin/pacbioget?11602). The characteristics of the identified motifs are shown in Table 1.

All subsequent calculations for motif characteristics were performed using in-house R and Python scripts.

The genomes of WT *M. gallisepticum* and ΔS.MgaS6I mutants were compared using Mauve [39].

The circular genome was visualized using Circos [40].

Specificity subunits with similar specificity to S.MgalS6I are described in S8 Table in S1 File. TRDs were predicted from HsdS protein sequences as those between long conservative alpha helices [21, 25]. The secondary structure prediction for HsdS was built using JPred4 [41]. TRDs with similar specificity for recognizing AGG or ANC half motifs were found in the REBASE database [7]. Alignments of TRDs were made using Clustal Omega [42]. The phylogenetic tree of TRDs was constructed using the Neighbor-Joining tree algorithm and visualized using Jalview 2 [43], and distance measures were calculated using BLOSUM62 substitution matrices.

## RNA isolation and RT-qPCR

RNA was isolated from *M. gallisepticum* and its transformants in the exponential phase of growth using Trizol® LS (Thermo Fisher Scientific), according to the manufacturer's instructions. The amount of RNA was determined using a Qubit® 2.0 fluorometer (Thermo Fisher Scientific), and the quality of RNA was estimated by conducting 1.5% agarose gel electrophoresis. RNA was treated with DNase I (Thermo Fisher Scientific). cDNA was synthesized from

random hexamer primers using Maxima™ H Minus Reverse Transcriptase (Thermo Scientific). Real-time PCR was performed using dNTPs, PCR buffer, Taq polymerase (Lytech, Russia), SYBR® Green I (Invitrogen, USA), and a CFX96 Real-Time PCR Detection System (Bio-Rad, USA), as described previously [37]. The primers used for *EGFP* were EGFP_F 5′-GACCCT GAAGTTCATCTGCACC-3′ and EGFP_R 5′-TAGTTGTACTCCAGCTTGTGCC-3′; topA_F 5′-AACTTGAAGAAGTCAGTGGGAT-3′ and topA_R 5′-TCTTACCATATTCTGGACTAA GACG-3′; primers for other genes have been described previously [37]. Primers were designed using BAC-Browser tools [44] and NCBI Primer-BLAST [45]. Quantitative data were normalized to the geometric mean of the *eno*, *gapd*, *tuf*, *gyrB*, and *ligA* genes. Relative changes in gene expression were calculated using the $2^{-\Delta\Delta CT}$ method. The significance of changes was calculated using Student's *t*-test with multiple hypotheses testing correction using the Benjamini–Hochberg procedure.

## 2D-DIGE

Cell lysis, 2D-DIGE, and sample trypsin digestion with subsequent MALDI analysis were performed as previously described [2]. After cell lysis, proteins were covalently tagged with CyDyeDIGE Cy5 (red, WT *M. gallisepticum*) and CyDye DIGE Cy3 (green, knockout mutant) (Amersham Bioscience, USA), according to the manufacturer's instructions (400 pmol for 50 μg protein).

To assess the effects of tetracycline or tetracycline resistance on protein abundance, we performed control 2D-DIGE with three other knockout mutants with *tetM* insertion in the GCW_01045, GCW_01060, and GCW_03935 genes.

## Protective function of the MgaS6I RM system

Four strains with *tetM* insertion in various intergenic regions were selected for comparison from the same library of random transposon knockouts (Arzamasov et al., manuscript in preparation) as the strain ΔS.MgaS6I. Construction of a vector for transformation of *M. gallisepticum* was performed as described in [3] except for the presence of the chloramphenicol resistance gene *camR* instead of the tetracycline resistance gene *tetM*. All strains before transformation were in the middle-log phase. Transformation was performed by means of electroporation, as described previously [38]. Selection of clones with *camR* insertion was carried out by growing the cells in the presence of 20 mkg/ml chloramphenicol. CFUs obtained after transformation were normalized to the mass of total RNA.

## Supporting information

**S1 Fig. Genome alignment of Mycoplasma gallisepticum strains obtained using the program Mauve.** RM system MgaS6I is marked in orange, variable lipoproteins vlhA clusters in green, the CRISPR system in blue. *M. gallisepticum* S6 genome is on the top, below the genomes of the other strains. A–whole genome representation, B–fragment of genomes near MgaS6I genomic context.
(PDF)

**S1 File. Supplementary tables S1-S8.**
(XLSX)

## Acknowledgments

We thank Mr. Lars Paulin and DNA Sequencing and Genomics Lab, Institute of Biotechnology, University of Helsinki for the PacBio sequencing.

## Author Contributions

**Conceptualization:** Tatiana A. Semashko, Gleb Y. Fisunov, Vadim M. Govorun.

**Investigation:** Tatiana A. Semashko, Alexander A. Arzamasov, Daria V. Evsyutina, Gleb Y. Fisunov.

**Methodology:** Tatiana A. Semashko, Alexander A. Arzamasov, Daria V. Evsyutina, Daria S. Matyushkina, Valentina G. Ladygina, Olga V. Pobeguts, Gleb Y. Fisunov.

**Software:** Tatiana A. Semashko, Daria V. Evsyutina, Irina A. Garanina.

**Supervision:** Vadim M. Govorun.

**Visualization:** Tatiana A. Semashko.

**Writing – original draft:** Tatiana A. Semashko.

**Writing – review & editing:** Alexander A. Arzamasov, Daria V. Evsyutina, Gleb Y. Fisunov, Vadim M. Govorun.

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
