## [Decision Letter · Decision Letter 0]

17 Aug 2022

PONE-D-22-17373Role of DNA modifications in Mycoplasma gallisepticum

PLOS ONE

Dear Dr. Semashko,

Thank you for submitting your manuscript to PLOS ONE. After careful consideration, we feel that it has merit but does not fully meet PLOS ONE’s publication criteria as it currently stands. Therefore, we invite you to submit a revised version of the manuscript that addresses the points raised during the review process.

1.  Please address comments made by the reviewer and myself.  Most comments ask for clarification of statements made in the manuscript. 2.  Reviewer 1 requests additional informatics work to address the probability of finding a methylation motif in a promoter region normally, given the GC content of the organism.  Because this information is important to support arguments made in the manuscript, this analysis should be performed and the results included in your revised manuscript. 3. Include additional evidence to support your suggestion that the RM system was acquired by horizontal gene transfer in your revised manuscript.

We look forward to receiving your revised manuscript.

Kind regards,

Mary Bryk, Ph.D.

Academic Editor

PLOS ONE

Journal Requirements:

  "This work was supported by the Russian Science Foundation grant no. 19-74-10105 “The role of chromatin structure in minimal cell in the maintenance of housekeeping proteome homeostasis.”"

Additional Editor Comments:

This is a solid manuscript that is well written. I identified a few points that require clarification.

Lines 176-177. The m6A modification and DNA modification motif ANCNNNNCCT were identified only in the WT strain, and the ΔS.MgaS6I strain lacked them. I do not understand why the DNA modification motif ANCNNNNCCT was not identified in the ΔS.MgaS6I strain.

I did not understand the meaning of the statement in lines 177-178, "This indicates that the RM system common for M. gallisepticum strains is nonfunctional, unless in the S6 strain." Are you suggesting that the common RM system is functional in the presence of the unique RM system in MgaS6I?

Reviewers' comments:

Reviewer's Responses to Questions

**Comments to the Author**

1. Is the manuscript technically sound, and do the data support the conclusions?

Reviewer #1: Yes

2. Has the statistical analysis been performed appropriately and rigorously? 

Reviewer #1: Yes

3. Have the authors made all data underlying the findings in their manuscript fully available?

Reviewer #1: Yes

4. Is the manuscript presented in an intelligible fashion and written in standard English?

Reviewer #1: Yes

5. Review Comments to the Author

Reviewer #1: In this work, the authors investigate genome methylation within Mycoplasma gallisepticum. The work itself is mostly solid (though I take issue with the “underrepresentation” part), so it’s rather unfortunate that most of the results are negative. The authors find one methylation motif and associate it with the one predicted functional RM system in the genome. They show this RM system basically does what RM systems are known to do, and the system has no other real impact on the physiology of the organism. It’s not transformative research, but I can’t blame the authors for that.

Major comments

It is my understanding that SMRT sequencing only is able to detect m6A modifications on native DNA, and special procedures have to be performed to identify m5C methylations. The authors should comment on whether they performed these procedures, otherwise we don’t know if they missed m5C modifications or not.

One of the pieces of data the authors present is the general lack of methylation sites within promoter regions. However, it’s not clear if that has occurred randomly or it has been selected for. I would have liked to have seen some informatic work to give an indication on what the probability of finding a methylation motif in a promoter region would be normally, given the GC content of the organism. Is the actual occurrence less than expected by random chance? Or is it that the genome has reduced so much and the motif itself is not simple enough where just randomly the motif doesn’t occur in those spots. Basically, I want to know if nature has selected for removal of methylation motifs from promoters because it could be disruptive. Without this analysis the authors cannot state that methylation sites are “underrepresented” in these places as they do later in the manuscript. They could only say they were not detected. We need to know what the actual expected representation would be to know if the sites are actually underrepresented.

As a side note, I appreciate that the authors did not try to over-extend their transcriptional change data. It would have been easy to present some of those transcriptional changes as significant in the figure, but they relied on the statistics and showed that most genes showed no significant change in gene expression. That’s good science.

Line 320 – the authors state the RM system was acquired by horizontal gene transfer. That’s a big statement to make. I would make sure to provide all the evidence that supports it. The only thing that comes to mind right now is that it’s present in the S6 strain and none of the others. But there are things that could be done to back up that statement.

Line 328 – honestly, I just don’t understand what the authors are talking about in this paragraph. Please clarify.

Minor comments:

Line 16-17 – poor grammar, should be “investigated DNA modifications of the model”

Line 178-179 – I’m confused by this sentence. The authors state that the RM system common to the different strains of M. gallisepticum is non-functional, which makes sense based on the mutations in this system, but then include the clause “unless in the S6 strain”. It’s this last part I don’t understand. I thought the point was that the unique system is responsible for the methylation, but that clause indicates the common system is responsible for the methylation.

Line 195 – I believe there is a typo, and it should be MgaS6I instead of MhaS6I

6. PLOS authors have the option to publish the peer review history of their article (what does this mean?). If published, this will include your full peer review and any attached files.

Reviewer #1: No

---

## [Author Response · Author response to Decision Letter 0]

15 Oct 2022

Dear Editor and Reviewer!

Please accept the revised manuscript «Role of DNA modifications in Mycoplasma gallisepticum» of the following authors: Tatiana Semashko, Alexander Arzamasov, Daria Evsyutina, Irina Garanina, Daria Matyushkina, Valentina Ladygina, Olga Pobeguts, Gleb Fisunov and Vadim Govorun for consideration as a publication in PLOS ONE journal.

Thank you for your time and for your appreciation of our work. We have corrected our manuscript in accordance with all your comments and suggestions. We hope you find our manuscript suitable for publication.

Response to Editor.

1. Please address comments made by the reviewer and myself. Most comments ask for clarification of statements made in the manuscript. 

2. Reviewer 1 requests additional informatics work to address the probability of finding a methylation motif in a promoter region normally, given the GC content of the organism. Because this information is important to support arguments made in the manuscript, this analysis should be performed and the results included in your revised manuscript. 

We have performed the requested calculations and corrected the text of the manuscript in accordance with results.

3. Include additional evidence to support your suggestion that the RM system was acquired by horizontal gene transfer in your revised manuscript.

We included supplementary figure with comparing genomes of different strains, extended paragraph “RM systems in M. gallisepticum and their functionality” in the Results and extended the reasons of our suggestion that the RM system was acquired by horizontal gene transfer in the Discussion.

Additional Editor Comments:

Lines 176-177. The m6A modification and DNA modification motif ANCNNNNCCT were identified only in the WT strain, and the ΔS.MgaS6I strain lacked them. I do not understand why the DNA modification motif ANCNNNNCCT was not identified in the ΔS.MgaS6I strain.

I did not understand the meaning of the statement in lines 177-178, "This indicates that the RM system common for M. gallisepticum strains is nonfunctional, unless in the S6 strain." Are you suggesting that the common RM system is functional in the presence of the unique RM system in MgaS6I?

Thank you for your comments. Both these statements are the results of mistakes. We fixed them.

Response to Reviewer.

Major comments

1. It is my understanding that SMRT sequencing only is able to detect m6A modifications on native DNA, and special procedures have to be performed to identify m5C methylations. The authors should comment on whether they performed these procedures, otherwise we don’t know if they missed m5C modifications or not.

Yes, you are right. For identification of DNA modification using SMRT sequencing without additional special procedures, 25x sequencing fold coverage is required for the identification of 4-methylcytosine, glucosylated 5-hydroxymethylcytosine, 6-methyladenine and 8-oxoguanine; 250x sequencing fold coverage is required for the identification of 5-methylcytosine and 5-hydroxymethylcytosine [https://www.pacb.com/wp-content/uploads/2015/09/WP_Detecting_DNA_Base_Modifications_Using_SMRT_Sequencing.pdf]. In our study the coverage of the genome per strand was approximately 90× for WT M. gallisepticum S6 and approximately 110× for the ΔS.MgaS6I strain. We didn’t find annotated m5C-methyltransferases in M. gallisepticum genome and didn’t identify m5C modifications. But if m5C-methyltransferases were present, we would have seen a not very significant signal for m5C modifications in our experiment.

2. One of the pieces of data the authors present is the general lack of methylation sites within promoter regions. However, it’s not clear if that has occurred randomly or it has been selected for. I would have liked to have seen some informatic work to give an indication on what the probability of finding a methylation motif in a promoter region would be normally, given the GC content of the organism. Is the actual occurrence less than expected by random chance? Or is it that the genome has reduced so much and the motif itself is not simple enough where just randomly the motif doesn’t occur in those spots. Basically, I want to know if nature has selected for removal of methylation motifs from promoters because it could be disruptive. Without this analysis the authors cannot state that methylation sites are “underrepresented” in these places as they do later in the manuscript. They could only say they were not detected. We need to know what the actual expected representation would be to know if the sites are actually underrepresented.

We thank the reviewer for pointing this out. In previous version of manuscript we used two-tailed Fisher’s exact test to validate the observed number of ANCNNNNCCT motif in promoter’s regions compared to occurrence it across whole genome. The promoter sequences in M. gallisepticum S6 were determined in our previous work [Fisunov GY et. al. Reconstruction of Transcription Control Networks in Mollicutes by High-Throughput Identification of Promoters. Front Microbiol. 2016 Dec 6;7:1977]. Now according your suggestion we tried to estimate motif underrepresentation in the promoter regions through ratio of observed over expected motif counts. There are several methods to estimate the expected frequency of short oligonucleotides in the genome. The simplest ones are based on the frequencies of mono- or dinucleotides in the genome. More complex methods use the maximal order Markov model or method by Samuel Karlin [Karlin S, Cardon LR. Computational DNA sequence analysis. Annu Rev Microbiol. 1994;48:619-54.], [Elhai J. Determination of bias in the relative abundance of oligonucleotides in DNA sequences. J Comput Biol. 2001;8(2):151-75.]. We calculated ratio of observed over expected motif through frequencies of each nucleotide in M. gallisepticum S6 genome and promoter regions. For whole genome, ratio was 1.29; for promoter regions - 0.2. It means there is no ANCNNNNCCT avoidance in whole genome of M. gallisepticum S6, but it probably exists in promoter regions. We also estimated enrichment ratio of ANCNNNNCCT in set of promoter sequences and in 100 sets of random generated sequences with the same length and GC content of the M. gallisepticum S6. In this case enrichment ratio was 0.49±0.11.

3. As a side note, I appreciate that the authors did not try to over-extend their transcriptional change data. It would have been easy to present some of those transcriptional changes as significant in the figure, but they relied on the statistics and showed that most genes showed no significant change in gene expression. That’s good science.

Thank you!

4. Line 320 – the authors state the RM system was acquired by horizontal gene transfer. That’s a big statement to make. I would make sure to provide all the evidence that supports it. The only thing that comes to mind right now is that it’s present in the S6 strain and none of the others. But there are things that could be done to back up that statement.

The MgaS6I system does not have close homologs in other M. gallisepticum strains. Additionally, it has a unique location in a genomic context compared to other strains. We added figure S1 with alignments of genomes of M. gallisepticum different strains. We extended of arguments for our suggestion that the RM system was acquired by horizontal gene transfer in text of the manuscript: in the paragraph “RM systems in M. gallisepticum and their functionality” in the Results and in the second paragraph of the Discussion.

5. Line 328 – honestly, I just don’t understand what the authors are talking about in this paragraph. Please clarify.

In this paragraph we discuss underrepresentation of recognition sites in whole genome of M. gallisepticum S6. Sites are not underrepresented comparing with genomes of other strains. We have rewritten this paragraph.

Minor comments:

Line 16-17 – poor grammar, should be “investigated DNA modifications of the model”

Thank you, we fixed it.

Line 178-179 – I’m confused by this sentence. The authors state that the RM system common to the different strains of M. gallisepticum is non-functional, which makes sense based on the mutations in this system, but then include the clause “unless in the S6 strain”. It’s this last part I don’t understand. I thought the point was that the unique system is responsible for the methylation, but that clause indicates the common system is responsible for the methylation.

Thank you, we fixed this typo.

Line 195 – I believe there is a typo, and it should be MgaS6I instead of MhaS6I

Thank you, we fixed it.

Best regards,

Tatiana Semashko

Address: Nauchnyj proezd 18, Moscow, Russia

Email: t.semashko@gmail.com

---

## [Editor Report · Decision Letter 1]

4 Nov 2022

Role of DNA modifications in Mycoplasma gallisepticum

PONE-D-22-17373R1

Dear Dr. Semashko,

We’re pleased to inform you that your manuscript has been judged scientifically suitable for publication and will be formally accepted for publication once it meets all outstanding technical requirements.

Kind regards,

Mary Bryk, Ph.D.

Academic Editor

PLOS ONE

Additional Editor Comments (optional):

Thank you for addressing the major and minor comments thoughtfully. I found three minor errors that you will want to correct.

Line 216 “was” should be “were”.

Line 361 “of its own operon” should be “in its own operon”.

Line 388 “this data” should be “these data”.

---

## [Editor Report · Acceptance letter]

14 Nov 2022

PONE-D-22-17373R1 

Role of DNA modifications in *Mycoplasma gallisepticum*

Dear Dr. Semashko:

I'm pleased to inform you that your manuscript has been deemed suitable for publication in PLOS ONE. Congratulations! Your manuscript is now with our production department. 

Kind regards, 

on behalf of

Dr. Mary Bryk 

Academic Editor

PLOS ONE